# Recent Advancements in Natural Plant Colorants Used for Hair Dye Applications: A Review

**DOI:** 10.3390/molecules27228062

**Published:** 2022-11-20

**Authors:** Hongyan Cui, Wenjing Xie, Zhongjie Hua, Lihua Cao, Ziyi Xiong, Ying Tang, Zhiqin Yuan

**Affiliations:** 1Beijing Key Laboratory of Plant Resources Research and Development, College of Chemistry and Materials Engineering, Beijing Technology and Business University, Beijing 100048, China; 2State Key Laboratory of Chemical Resource Engineering, College of Chemistry, Beijing University of Chemical Technology, Beijing 100029, China

**Keywords:** plant hair dye, natural colorant, coloration mechanism, mordant, encapsulation, cosmetic safety

## Abstract

There is an on-going demand in recent years for safer and “greener” hair coloring agents with the global consumer awareness of the adverse effects of synthetic hair dyes. The belief in sustainability and health benefits has focused the attention of the scientific community towards natural colorants that serve to replace their synthetic toxic counterparts. This review article encompasses the historical applications of a vast array of natural plant hair dyes and summarizes the possible coloration mechanisms (direct dyeing and mordant dyeing). Current information on phytochemicals (quinones, tannins, flavonoids, indigo, curcuminoids and carotenoids) used for hair dyeing are summarized, including their botanical sources, color chemistry and biological/toxicological activities. A particular focus is given on research into new natural hair dye sources along with eco-friendly, robust and cost-effective technologies for their processing and applications, such as the synthetic biology approach for colorant production, encapsulation techniques for stabilization and the development of inorganic nanocarriers. In addition, innovative in vitro approaches for the toxicological assessments of natural hair dye cosmetics are highlighted.

## 1. Introduction

Nowadays, with the growing global awareness of the adverse effects of synthetic hair dyes, the demand for safer and more environmentally friendly hair dyes is increasing. Hair dye products can be grouped into three categories according to wash fastness: temporary, semi-permanent and permanent hair dyes [1]. Permanent hair dyes refer to synthetic oxidative hair dyes, by which colors are produced in the hair cortex from small primary intermediates (e.g., *p*-phenylenediamine and *p*-aminophenol) and couplers (e.g., *m*-aminophenol, *m*-hydroxyphenol and resorcinol) through oxidation reactions in the presence of hydrogen peroxide as the oxidizing agent [2]. Permanent hair dyes represent the most widely used coloring matter in commercial hair dye cosmetics due to their strong dyeing performance, predictable colors and rich range of tones [3]. However, several studies have reported allergenicity [4,5], mutagenicity [6,7], carcinogenicity [8], and environmental toxicity [9] associated with the use of synthetic hair dye ingredients and the potential health risks have attracted widespread attention. By contrast, natural dyes are temporary or semi-permanent non-oxidative hair dyes that can be adsorbed onto the cuticle and some parts of the cortex of the hair shaft to produce color. Natural dyes derived from various parts of plants (e.g., fruits, flowers, leaves, seeds and roots) are generally regarded as low-irritating, less allergenic, sustainable and eco-friendly green products with additional health benefits (e.g., antioxidant, anti-inflammatory and antimicrobial properties) [10].

Natural dyes have been used since ancient times, when they were used not only for hair coloration, but also for medicinal, decoration and religious purposes [11,12]. In the early days, hair dyes were obtained from metallic compounds, plant extracts, dried plants or their mixtures [13]. Before the invention of first synthetic aniline dye, mauve, in 1856, different plant extracts and herbal preparations such as mullein, birch bark, turmeric, and saffron have been used for hair dyeing. The early record of natural hair dyeing dates back to ancient Egyptian times when Rameses II reinforced his red hair color using henna [13]. The ancient Greeks used to bleach their hairs using a rinse of potassium lye solution followed by rubbing with a type of ointment made of yellow flower petals and pollen [14]. The Romans dyed their hair black by using walnut extracts [15]. Today, the renascence of natural botanical ingredients in cosmetics and health care products has led to research work into the phytochemistry and coloring potential of these traditionally used hair dye plants. Compounds including quinones, tannins, flavonoids, indigo, curcuminoids and carotenoids were identified as the dominant naturally-occurring hair coloring matters and some plants accumulating these phytochemicals, such as *Lawsonia inermis* (henna) [16,17,18], *Juglans regia* (walnut) [11,12,16], *Curcuma longa* (turmeric) [19,20], *Haematoxylon campechianum* (logwood) [16,19,21] were extensively investigated. Natural dyes used in commercial cosmetics are mainly extracted from plants by solvent extraction [19], ultrasonic assisted extraction [22], microwave assisted extraction [10], supercritical fluid extraction [23], and enzyme-assisted extraction [24] etc. 

This review focuses on complete phytochemical information of plant-derived hair dyes including historical uses, coloring mechanisms, color chemistry, and toxicological aspects. The development of state-of-the-art techniques to produce or stabilize natural plant dyes for advanced hair dye applications as well as hair dye safety assessments are also highlighted. The purpose of this review is to facilitate further exploration of natural colorants as healthy and sustainable hair dye products.

## 2. Hair Coloring Mechanisms

For most natural plant hair dyes, there are two mechanisms for hair coloration: direct dyeing and mordant dyeing. Briefly, the hair dyeing process can be divided into two steps: (i) Diffusion of dye molecules from dye bath to the keratinous hair fiber; (ii) Formation of chemical bonds (hydrogen, ionic, and covalent bonds) between the carboxyl or hydroxyl groups present in the dye molecules and amino/sulfhydryl groups in hair keratin, with or without the aid of auxiliary mordanting agents [25].

Diffusion is a three-stage process [1]: (i) the first stage is the transport of dye molecules to the fiber/water interface by a combination of aqueous diffusion and agitation; (ii) in the second stage, dyestuffs are adsorbed onto the outer layer of hair cuticle; (iii) final stage is the diffusion of dye molecules of low molecular weight into inner hair structures (cuticle and cortex) and can be characterized by the change to the cell membrane complex (CMC) present in the hair cuticle. The CMC is a continuous phase of intercellular matters that binds the cuticle and cortical cells together [26]. Studies have shown that penetration though CMC is the main transport pathway for dye substances to reach the hair cortex [27,28]. Less ionized small molecules are more likely to penetrate through and spread over the lipid bilayer of CMC [29]. Besides, the condition of hair fibers also affects the absorption and diffusion of external dye substances. For example, the use of hydrogen peroxide in hair dye formulation can destroy the disulfide bonds of hair keratin, causing CMC breakage and damages to the cuticle and cortex components, resulting in swelling loose hair fibers and lifted cuticles, thus facilitating deeper penetration and stronger bonding of dyestuffs to the hair exterior shaft [12,19].

Direct dyeing, as a non-oxidative hair coloring process, is a direct formation of a dye-complex or bonding between the dyestuff and hair fiber. The color strength of directly dyed hairs depends on the affinity of dye molecules to the hair fiber surface. Generally, dyestuffs of low molecular weight (the critical sizes are 1.2–1.3 nm for anionic dyes, 1.4 nm for cationic dyes, and 0.95 nm for nonionic dyes [30]) can easily penetrate into the cuticle layer of hair fiber. Dyes of high molecular weight cannot penetrate the cuticles but may be adsorbed onto hair fiber via various types of forces, i.e., van der Waals, electrostatic, and hydrogen bonding [31]. Among the direct dyes, natural dyes extracted from henna leaves and walnut husks are popular representatives. Take henna for example, at pH 4.5–6.0, the reduced form of lawsone (2-hydroxy-1,4-naphthoquinone), its main colorant, reacts with the protonated amino groups present in hair keratin fibers (Figure 1a) [32]. Additionally, SEM observation finds that henna dyestuffs might be capable of recovering the cuticle damage and providing a smooth moisture-rich appearance on the dyed hair cuticles [33].

Mordant dyeing refers to the formation of charge-transfer complex between the dye and a mordanting agent on the dyed hair [34]. Mordants are substances that can fix dyes on hair fibers through interactions with dye molecules and hair fibers for improved color fastness [35]. Common mordants are metal salts, such as iron (II) sulfate, copper (II) sulfate, and alum, and act as a link between the dye and the hair fiber. Dative covalent bonding is the probable mechanism in binding metallic mordants to dye molecules with the oxygen-containing groups playing a key role [19]. Figure 1b shows dative covalent bonding between an iron (II) ion and a polyphenol dye that bonds to the hair fiber by hydrogen bonding [36]. Mordant dyeing can be conducted by pre-, meta- or post-mordanting methods. The choice of mordants and mordanting methods has significant influence on the hue of the dyed colors and fastness properties [37]. For natural dyes, it is difficult to predict an optimal mordanting procedure because the results are highly dependent on the dye plant and mordant type. On the other hand, the treatment of transition metal mordants may result in the accumulation of iron and copper in human hairs, which has been reported to cause photooxidative damage of dyed hairs through Fenton chemistry [38]. In this regard, the development of bio-mordants, especially tannins and metal-rich plants, as effective alternatives to metallic mordants warrants further investigation [39]. For example, *Aloe vera* extract was reported as a bio-mordant to improve the hair dyeing properties [40]. Tannin-rich plant extracts from *Punica granatum* (pomegranate) peels, *Eucalyptus maculata* (eucalyptus), *Rhus coriaria* (sumac) and *Emblica officinalis* (amla) are bio-mordants widely used in the textile industry as alternatives to the metallic mordants but are rarely reported for hair dye applications [41].

**Figure 1 molecules-27-08062-f001:**
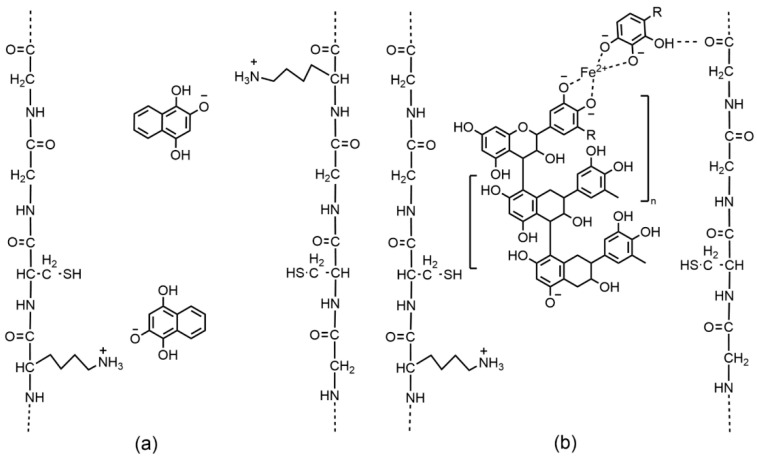
Possible interactions between hair fibers and lawsone (**a**); Possible interactions between hair fiber, polyphenol dye and mordanting iron (II) ions, where R=H, CH_3_, OH (**b**). Reprinted with permission from Ref. [36]. Copyright 2021, Multidisciplinary Digital Publishing Institute.

## 3. Phytochemicals Used for Hair Dyeing

Many organic compounds have been identified as the principal coloring matters in hair dye plants and investigated for dyeing performance under experimental conditions. Natural colorants can be classified based on dye source, application method and chemical structure. Here we describe the structure-based classification since the chemical structure uniquely identifies dye molecules with specific properties (Table 1). 

### 3.1. Quinones

Quinones are colored compounds with a basic benzoquinone chromophore consisting of two carbonyl groups. The three main classes of quinones are naphthoquinones, anthraquinones and benzoquinones [53]. Among which, naphthoquinones, widely distributed in plants and microorganisms, are the most frequently encountered quinone hair dyes. In plants, these compounds usually exist in the free form with several isomers, among which, 1,4-naphthoquinones are the most stable [54]. The light absorbance of quinone dyes depends on their skeleton structure and is affected by the presence of various substituents. The introduction of substituents, especially free or methylated hydroxyl groups, may induce a red shift of the absorption maxima. Some substituents, such as amino or substituted amino groups, may have significant influences on the color properties of quinone dyestuffs [55]. Representative naphthoquinones for hair dyeing purposes are lawsone, juglone, and shikonin (Figure 2). 

Lawsone (2-hydroxy-1,4-naphthoquinone) is the main coloring component of henna leaves, also known as CI Natural Orange 6, which acts as a direct or mordant dye for hair fiber to give a red-orange color [17]. In Asia and North Africa, the leaves of henna are widely used for hair and skin coloration. In a traditional manner, henna paste (powdered henna leaves mixed with warm water) is directly applied onto the hair where lawsone can gradually diffuse from the paste into the hair shaft to bind with keratin. There are many natural dye formulations based on henna on the market, usually in the powder form. For example, a gel formula consisting of powdered henna, tea, and hibiscus leaves can dye bleached hair brown [18]. Besides direct dyeing, henna can also be used with various mordants. Ali et al. [20] reported that the plant hair dyes prepared from henna, curcumin, and *Tagetes erecta* extracts can be used as both direct dye to color gray hair brown and as mordant dye to obtain black color when combined with iron (II) sulfate. Additionally, lawsone has a very low allergic potential. In most cases, allergic reactions are not caused by the henna itself, but by the synthetic coloring additives that are added to henna mixtures [56]. It is worth noting that, when pure henna (usually considered as a weak sensitizer) is used in combination with *p*-phenylenediamine, the risks for inducing sensitization and broad immune responses (including hypersensitivity) increase significantly [57]. The European Union Scientific Committee on Consumer Safety (SCCS) concluded that henna was slightly and transiently irritating to the eye but it can be safely used as a hair dye when the content of lawsone is less than 1.4% [58]. Besides lawsone, there are flavonoids, tannins, phenolic compounds, alkaloids and other active components in henna. Researchers have reported that henna extract has anti-inflammatory, anticonvulsant [59], antioxidant, immunomodulatory [60], wound healing [61,62], and other pharmacological activities. 

Juglone (5-hydroxy-1,4-naphthoquinone) is an isomer of lawsone and can be obtained from the leaves, roots, shells, and barks of walnut plants. Juglone and juglone-containing walnut green husk extracts were used in skin coloring preparations [63], hair dyes [11,12,16] and antimicrobial agents [64]. As a hair dye, juglone gives hair a brownish color when mordanted with iron (II) ions. Beiki et al. reported the use of walnut husk extract to dye bleached hair with good washability and antibacterial activity [12]. A variety of in vitro test methods were used to evaluate the safety of plant hair dyes based on walnut husk extract, categorizing it as non-irritant to the skin but slightly irritating to the eye [16]. Additionally, some studies reported dose-dependent cytotoxicity of juglone in human fibroblasts and keratinocytes [65,66]. These findings indicate that cosmetic preparations containing juglone should be used with care. 

Likewise, shikonin has also been used as a direct hair dye for brown color since ancient times. Shikonin ((R)-5,8-dihydroxy-2-(1-hydroxy-4-methylpent-3-en-1-yl)naphthalene-1,4-dione) is a dominant component in the roots of *Lithospermum erythrorhizon* (gromwell root), a perennial herbaceous plant native to China, Japan, and Korea [67,68]. Wang et al. reported that the dyeing performance of gromwell root extracts on bleached hairs with the most pronounced color change observed under acidic conditions [21]. Besides, shikonin is believed to endow fibers with antibacterial and anti-ultraviolet properties [69].

Anthraquinones are compounds with a central 1,4-diketo-cyclohexa-2,5-diene (quinone) structure connected to peripheral phenyl rings [70]. Anthraquinone dyes are typical donor/acceptor types and the substituent effects of which can provide a wide range of colors. A notable structural feature is the hydrogen bonding between α-substituents and the carbonyl group, which enhances light fastness [34]. Anthraquinones are found in various parts of plants (flowers, leaves, fruits, roots, and rhizomes). *Rheum officinale* (rhubarb), *Hamamelis mollis* (witch hazel), *Aloe vera* (aloe), and *Rhamnus davurica* (buckthorn) are known genera rich in these compounds [71]. In plants, anthraquinones usually exist in the form of glycosides and rarely in the free form. *Rubia cordifolia* (madder) is a common source of the anthraquinone dye alizarin for dyeing purposes [53]. The structure and color of alizarin are pH-dependent (Figure 2). When the pH value is less than 7, alizarin is in its unionized form with both phenolic hydroxyl groups closed and appears yellow with the absorbance maximum at 430 nm [72,73]. At weak alkaline pH 8, alizarin is partially deprotonated only with its β-OH and appears red with two absorption peaks at 430 nm and 530 nm [73]. When pH increases (8 < pH < 13), both phenolic hydroxyl groups lost hydrogen, and the absorption peak of alizarin moves rapidly to 530 nm, showing a purple color [74]. Boga et al. investigated hair dyeing with alizarin at pH 8 (using yak hair as a model) and found it can rapidly turn white yak hair into reddish color [11].

Benzoquinone is the basic subunit of quinone compounds. 1,4-benzoquinone is a low molecular weight benzoquinone dye extracted from young shoots of the *Pyrus lindleyi* (pear) and can be used for direct dyeing [53]. It was reported that 1,4-benzoquinone can give yak hairs a brown color at pH 2.7 or pH 8 in a concentration dependent manner, and the color intensity does not seem to be strongly affected by pH [11]. 

Overall, quinones are a class of naturally abundant colorants with great potential to obtain a broad spectrum of colors on hair, ranging from deep purple to orange and yellow. Currently, the application of quinone colorants in commercial hair cosmetics is limited by poor solubility, strong odor, and photodegradation susceptibility.

### 3.2. Tannins

Tannins are a large group of polyphenolic molecules having several phenolic hydroxyl groups and other groups such as carbonyls to form strong complexes with various macromolecules [75] (Figure 3). Tannins are broadly distributed in the plant kingdom and are the most abundant secondary metabolites [76]. They are generally classified into two types: hydrolysable tannins and condensed tannins [77]. Hydrolysable tannins contain glucose or polyhydric alcohols esterified with gallic acid (e.g., gallotannins) or hexahydroxydiphenic acid (e.g., ellagitannins) while condensed tannins consist of flavolans or polymeric parts of flavan-3-ols (catechins) and/or flavan 3:4-diols (leucoanthocyanidins) [75].

Tannins are typical mordant dyes. One famous example used in history is the iron-gallink (tannin-iron coordination compounds) [78]. When hydrolysable tannins (pale yellow color) are complexed with iron (II) ions, a strong bluish-black color is produced and the coordination complex formed between tannin, mordant and hair fiber can enhance the color fastness [16]. Both gallotannins and gallic acids occur abundantly (tannin content over 65%) in *Galla Chinensis* (Chinese gallnut) and the gallnut extract was widely used as a black hair dye with iron (II) sulfate in eastern Asia [44]. Sargsyan et al. [36] studied the hair dyeing mechanism of matcha, which contains another type of hydrolysable tannin, catechin, and found a similar iron-gall complex formed between the mordanting iron (II) ions and the hydroxy groups of flavanol and keratinous hair fiber [79]. Additionally, the selection of metal ions and mordanting methods has shown to have significant effects on the color intensity and fastness of tannin dyeing [80].

On the other hand, tannins can also act as a bio-mordant to enhance the dyeing fastness of many plant dyes. Jahangiri et al. [81] investigated the effects of tannin-based bio-mordants and metallic mordant (alum) on wool dyed with madder root extract. The results showed that fibers pretreated with tannin produced very similar color and washing fastness with those pre-mordanted with alum (ΔE < 1). This was suggested to result from the formation of ionic complexes among protein fibers and organic molecules with ionizable groups at appropriate pHs [82]. In addition, due to the existence of abundant −OH groups, the application of tannins as mordants may also enhance color fastness by forming additional hydrogen bonds with both colorants and protein fibers [83]. 

### 3.3. Flavonoids

Flavonoids are formed in plants from the aromatic amino acids, i.e., phenylalanine and tyrosine, and generally occur as glycosylated derivatives. The core structure of flavonoid is flavan nucleus, which consists of 15 carbon atoms arranged in three rings [84]. Flavonoids known for hair dye applications include anthocyanins, hematoxylin, quercetin, acacetin, etc. (Figure 4). Anthocyanins are the largest group of polyphenols in the plant kingdom. They are responsible for the pink, red, purple, violet and blue colors of many fruits, vegetables and flowers [19]. Several studies have demonstrated the successful use of plants containing anthocyanins for hair coloring, including *Ribes nigrum* (blackcurrant) [45], *Morus nigra* (mulberry) [11], *Phaseolus mungo* (bleak bean) [46] and *Cleistocalyx nervosum* var. paniala [47]. Their colors are determined by the number of hydroxyl groups and degree of methylation as well as the number and position of sugar moieties (glycosides) and attached aliphatic or aromatic acids [45]. Six common derivatives of anthocyanins are presented in Figure 4. Their colors are highly influenced by the environmental acidity/alkalinity. At pH < 3, the flavan nucleus exists mainly as flavylium cation (AH^+^) showing a red color [45]. When pH increases, AH^+^ undergoes a rapid deprotonation to form a purple-colored quinonoidal base (A); and when pH > 7.5, an anionic quinonoidal base (A^−^) is formed with a blue color [85]. Besides, opening of the anthocyanin ring may result in the formation of a yellow-colored E-chalcone [86]. Cyanidin-3-glucoside was successfully applied to hair coloring as a source of red colorants and can change into blue color when complexed with iron (II) oxalate [45,47]. Therefore, anthocyanin dyes can be used for both direct dyeing and mordant dyeing. However, anthocyanins are unstable in aqueous formula whereas the addition of vitamin E acetate at 0.04% can enhance this natural colorant’s stability [45]. Anthocyanins-based preparations have shown to be efficient semi-permanent dyestuffs for hair with the dyed colors durable up to 5 wash cycles [46]. 

Hematoxylin, (6aR,11bS)-7,11b-dihydroindeno[2,1-c]chromene-3,4,6a,9,10(6H)-pentaol, derived from *Haematoxylon campechianum* (logwood) [87], can act directly or as mordant dye for hair fiber to give a reddish-brown color [16]. Wang et al. reported the use of logwood extract to dye bleached hair and the dyeing effect was optimal at pH 7 [21]. Thermodynamic and kinetic studies have shown that the adsorption of hematoxylin on hair is a spontaneous and exothermic process [87]. In vitro toxicological tests demonstrated that logwood extract is a safe non-irritant hair dye ingredient [16].

Quercetin (2-(3,4-dihydroxyphenyl)-3,5,7-trihydroxy-4-Hchromen-4-one)) is a polyphenolic flavonoid found in tea, onions and berries [88] and can act as a direct dye for hair coloring. Tibkawin et al. reported the use of teak leaf extract (quercetin is the main colorant) as a plant hair dye for bleached human hairs wherein the color obtained is dependent on the harvest states of leaves (young leaf extract produces reddish brown while mature leaf extract produces brown color) [49]. Quercetin is also known for its medicinal bioactivities, such as anticancer and antiviral activities [89]

Acacetin (5,7-dihydroxy-4-methoxyflavone) is a naturally-occurring flavonoid in the bark of *Acacia farnesiana* (huizache) [90]. Ali et al. studied the use of huizache extract to dye gray hair, and the color obtained by direct dyeing was less intense than that dyed with henna extract [20]. Besides, acacetin has a variety of pharmacological properties including neuroprotective, cardioprotective, anticancer, anti-inflammatory, antidiabetic, and antimicrobial activities [91].

Collectively, flavonoid colorants may come from a wide range of sources, but their colors are easily affected by environmental pH, metal ions, light, temperature, and oxygen [92,93].

### 3.4. Indigo

Indigo, also known as indigotin (CI Vat Blue 1), has been used as a vat dye and traditional medicine for thousands of years. In ancient times, freshly picked stems and leaves of “daqingye” indigoid plants [94], i.e., *Indigo Naturalis*, *Baphicacanthus cusia (Nees)* Bremek., *Polygonum tinctorium* Ait. and *Isatis indigotica* Fort. were soaked in vat water for several days to ferment and become dark blue [95]. Thereafter, lime was added and the residues were stirred and precipitated to obtain indigo colorant, which is a glycosylated form of indole precursor (e.g. isatan A and B) [96]. During fermentation, indole precursors, i.e., hydrolyzed and released indoles, combine spontaneously to form indigo in the presence of oxygen [97]. Intra- and inter-molecular hydrogen bonding are responsible for indigo’s insolubility in water and dilute acid [98]. 

In the dyeing process (Figure 5), indigo is first chemically reduced in alkaline medium to obtain its soluble reduced form. Sodium hydrosulfite is usually used as the reducing agent for textiles dyed with vat dyes [32,99]. Indigo carmine (indigo-5,5′-disulfonic acid di-sodium salt, C.I. Natural Blue 2) was considered as sulfonated indigo which can give hair a blue hue when used as an acid dye [51]. In the case of hair dyeing, indigo is usually mixed with a certain proportion of henna powders and water to make a paste which can color the hair dark brown [50]. Komboonchoo et al. [32] investigated the dyeing characteristics of indigo, lawsone and lawsone-indigo mixture under reducing and/or oxidizing conditions and found that the dyeing performance of indigo dyes strongly depends on the pH of the solution. Indigo, when dyed in a strong alkaline dyeing bath, shows enhanced dye uptake with a dark blue color due to the presence of soluble monophenol and bisphenol ions of its reduced form. Besides, some studies have reported that indigo has anti-inflammatory, antioxidant, antibacterial and immune regulatory activities [100,101]. Overall, indigo is a sustainable and environmentally friendly natural colorant and is widely used in commercial hair dye cosmetics with henna for dark colors.

### 3.5. Curcuminoids

Curcuminoids are a bis-α, β-unsaturated diketone and in solution exist in equilibrium with the corresponding enol tautomer [11]. They are abundant in the rhizome of *Curcuma longa* (turmeric) and can be found in other Zingiberaceae and Araceae families. Under acidic and neutral conditions, the bis-keto form predominates, whereas at >pH 8 the enol tautomer is favored. Curcumin is practically water insoluble at acidic and neutral pHs while in alkali solution the formation of more soluble anionic species takes place [11]. Curcumin and other related curcuminoids (demethoxycurcumin and bisdemethoxycurcumin) are well known yellow curry colorants and textile dyes (Figure 6). It was reported that under acidic conditions, curcumin dispersed in a water/2-propanol/benzylalcohol solution can be used for direct hair dyeing, giving a distinct yellow color [11]. Curcumin can also be used for mordant dyeing. Under the action of iron (II) sulfate mordant, bleached hairs can be dyed into an orangish-brown color with resistance to 8 shampooing washes [19]. Besides, curcumin has been used in traditional medicine for treating diabetes, abdominal pains, menstrual disorders, wounds, eczema, jaundice, inflammations and for blood purification [52]. Curcumin extracted from turmeric was reported with potent antioxidant, anti-inflammatory, anticancer and hepatoprotective activities [102]. Nevertheless, poor water solubility and photostability limit the industrial use of curcuminoids in hair dye formations.

### 3.6. Carotenoids

Carotenoids are linear conjugated polyene-terranes and the general structure usually consists of a polyene chain with nine conjugated double bonds and two groups at both ends. Highly conjugated electronic systems of carotenoids contribute to their yellow, orange, red and purple colors [103]. Three types of carotenoids, i.e., zeaxanthin, peridinin, and lutein were successfully applied for hair dyeing (Figure 7). Carotenoid dyes can provide bright hues with good color fastness properties when associated with metallic mordants. Boonsong et al. reported the use of *Eclipta alba* (false daisy) extract (its main colorant is zeaxanthin) and *Terminalia belerica* (beleric myrobalan) extract (its main colorant is peridinin) as hair dye for bleached hair with ascorbic acid as a natural color developer and iron (II) sulfate as the mordant, which produced good dyeing performance and dye fastness [19]. Likewise, lutein obtained from *Tagetes erecta* (marigold) was used in natural hair dyeing to cover grey after mixing with *Cymphomandra betacea* (tamarillo) extract and *Aloe vera* mordant [40]. Besides, several carotenoids, such as bixin and norbixin, have been used as a good remedy for cardiac, astringent and febrifuge gonorrhea [102]. Similar to curcumin, carotenoids are hydrophobic dyes extracted using organic solvents such as hexane, acetone and ethyl acetate. Poor photostability is a hindrance to their applications in commercial cosmetic products.

## 4. Technological Innovations for Natural Hair Dyeing

### 4.1. Colorant Production by Synthetic Biology Techniques

Traditional methods of obtaining hair dye colorants from plants is limited by the dependence on complex extraction/purification procedures, varying botanical sources, long cultivation cycles and limited harvest seasons. In recent years, synthetic biology techniques have brought new tools for producing plant colorants by microbial fermentation. Metabolic and genetic engineering approaches have shown great potential to edit or introduce colorant-formation genes into microorganisms for enhanced production of anthocyanins, curcumin and carotenoids [104,105]. The first microbial synthesis of anthocyanins was reported by Yan et al. [106], wherein a four-step metabolic pathway containing heterologous plant genes was constructed in engineered *E. coli* and the cytosol was able to take up naringenin and mustard alcohol to produce anthocyanin 3-O-glucoside. Likewise, heterologous synthesis of curcumin was produced in engineered *E. coli* by using 4-coumarate-CoA ligase (4CL1) from *Arabidopsis thaliana*, diketide-CoA synthase (DCS) and curcumin synthase 1 (CURS1) from turmeric [107]. Additionally, a more efficient biosynthetic pathway of β-carotene was assembled in microorganisms by metabolic engineering techniques by increasing the copy numbers of the carB and carRP genes and overexpressing genes related to the mevalonate pathway [108]. However, despite the substantial advances in the fields of synthetic biology and metabolic engineering, the development of heterogenous microbial cell factories is cumbersome and time-consuming. Additionally, the potential biosafety risks in synthetic biology are a matter of great concern recently. Thus, it is still challenging for industrialization of such technologies to produce various types of natural colorants and apply them in hair dye cosmetics. 

### 4.2. Encapsulation of Colorants for Stabilization and Detoxification

Stability issues (thermal-, light and acidity/alkalinity stability) must be considered when using natural colorants in hair dye cosmetics. Many encapsulation systems have been successfully developed over the past years to protect sensitive phytochemicals or plant extracts from environmental stresses (heat, UV and extreme pHs) [109]. Encapsulation provides a promising strategy to protect sensitive dye molecules (core material) by enclosing them in an outer shell (wall material) to ensure the stability and dyeability of natural colorants in commercial hair dye products. Microcapsules of natural colorants can be obtained by various encapsulation techniques, including both chemical (emulsion polymerization, suspension polymerization, and interfacial polymerization) and physical (spray-drying, spray-cooling, and co-extrusion) approaches [110]. Tang et al. used maltodextrin and gum Arabic as wall materials to prepare microcapsules of extracted colorants from Chinese gallnut and henna by spray drying techniques. The encapsulation of gallnut extract significantly improved the photo- and thermal-stabilities as well as formulation stability in alkaline formulation while the encapsulation of henna extract remarkedly reduced its contact toxicity without affecting the hair dyeing properties [17,44]. Therefore, encapsulation technology has provided a route to solve the stability and compatibility issues addressed in natural dyeing with plant colorants. 

### 4.3. Development of Inorganic Nanocarriers for Efficient Hair Dyeing

Benefiting from small particle size, large surface area and nanostructure, and tailorable physicochemical properties, nanomaterials are ideal carriers of bioactive ingredients for cosmetic applications [111]. In recent years, various inorganic nano-carriers (e.g., nanoparticles, nanofibers and nanotubes) have shown usefulness in stabilizing hair dye plant dyes and enhancing the dyeing effect. For example, carbon nanotubes with small size and increased surface to volume ratio can easily be absorbed onto the hair cuiticles and interact with the hair fiber, leading to enhanced affinity and long-lasting coloring effects [112]. Similarly, gallic acid reduced/functionalized silver nanoparticles [113] and colored silica nanoparticles [114] have been developed as novel dyes for bleached human hairs owing to their local surface plasma resonance properties. Besides, oxidation of dopamine to eumelanin-like polydopamine and deposition on the surface of hair in the form of nanoparticles were shown as a novel biomimetic strategy to develop melanin-mimicking pigments for hair dyeing [115,116]. Furthermore, Panchal et al. [117] reported that halloysite nanotubes, when loaded with lawsone, were effective for the coloration of both pigmented and grey hairs with good shampooing fastness and can modify the hair surface through physical adsorption and self-assembly. There are also reports on hair dye applications of other abundant and easy-to-obtain natural clay minerals such as sepiolite, palygorskite and kaolin [118]. Collectively, inorganic nanomaterials incorporated/functionalized with hydrophilic or hydrophobic dye molecules from natural sources have provided possibilities for the development of novel natural colorants-based hair dye products. 

## 5. Toxicological Assessments

Natural origin does not necessarily mean non-toxic or safe. Black henna, a combination of henna leaves and *p*-phenylenediamine (PPD), is known to cause allergic contact dermatitis [56,119]. Some colorants, such as juglone, are natural toxins with cytotoxic effects [65,66]. Additionally, the colorant contents and toxic effects of plant extracts used in hair dye preparations can be variable with different plant sources, harvest seasons and extraction techniques. The application compatibility of juglone-containing walnut husk extract as a hair dye ingredient was assessed both in vitro and in vivo, showing it is not irritating to the skin [12,16]. Besides colorants, metallic mordants and heavy metal contaminants from plant extracts can be accumulated in humans with frequent dyeing and may impose health impacts. For example, the presence of high levels of Fenton transition metals (e.g., iron) can cause photooxidative damage of hair fiber [38]. Heavy metals, such as cadmium, chromium and lead, can be absorbed in the skin, liver and kidney, causing allergic contact dermatitis [120], reproductive system dysfunction [121], and other adverse health effects. Several novel analytical methods, such as differential pulse anodic stripping voltammetry, were developed for the rapid detection of heavy metals [122,123]. Therefore, safety assessments of both natural colorant ingredients and hair dye formulations are compulsory. It is also necessary to establish a toxicological database of hair dye plants and their main colorants since currently most plant extracts used in hair dye cosmetics are lacking important toxicological information such as acute/sub-chronic systemic toxicity, irritancy, allergenicity and genotoxicity [124]. 

Previously, cosmetic safety evaluations were mainly conducted using animal models [12,18]. Nowadays, with global acceptance of the 3R principles (replacement, reduction and refinement) and the EU ban on animal testing in 2013, more in vitro methods have been developed and applied in the toxicological assessments of plant hair dye ingredients and formula. Tang et al. evaluated the eye and skin irritation potentials of four plant hair dyes (henna, Chinese gallnuts, sappanwood, and walnut husks) by using several OECD validated in vitro methods, including bovine corneal opacity and permeability (BCOP) assay in combination with histopathological analysis, Hen’s egg test on chick chorioallantoic membrane (HET-CAM), and a test on reconstructed human epidermis models [16]. Likewise, a combination of in vitro methods, i.e., micro-direct peptide reactivity assay (mDPRA), HaCaT keratinocytes-associated IL-18 assay, U937 cell line activation test (USENS)/IL-8 levels, blood monocyte-derived dendritic cell test and genomic allergen rapid detection (GARD skin) were reported to assess the sensitization potential of henna-based hair dye products containing *p*-phenylenediamine [57]. 

## 6. Concluding Remarks

In recent years, green and environmental-friendly plant hair dyes have become the development trend of hair dye market owing to the growing awareness of consumer health. As an alternative to the synthetic hair dyes, botanical colorants are advantageous as they are green in nature, less toxic and biodegradable. Additionally, many plant colorants are known for their health-promoting bioactivities such as antioxidant, anti-inflammatory and antimicrobial properties. Natural colorants are advantageous for sustainability and potential health benefits while their industrial utilization for natural hair dyeing still needs to address the following challenges:Cumbersome extraction/purification procedures.High susceptibility to environmental pH, metal ions, UV and temperature.Low dye uptake and poor color fastness on hairs, especially unbleached hairs.Poor color reproducibility on human hair and testing models (e.g., yak hair and wool).Dependence on transition metallic mordant makes the dyed hair vulnerable to photo-oxidative damage and complicates the dyeing process.Insufficient toxicological data.

Future studies are required to understand the mechanistic interactions between various plant-derived colorants and human hairs, to provide a theoretical basis for the design of efficient plant hair dyes as well as to broaden the sources of hair dye plants or engineered microorganisms for large-scale production. Meanwhile, the development of efficient encapsulation systems, nanocarriers and bio-mordants, to improve the dyeability and color fastness of natural colorants as well as the establishment of toxicological databases for hair dye plants also warrant further investigation.

## Figures and Tables

**Figure 2 molecules-27-08062-f002:**
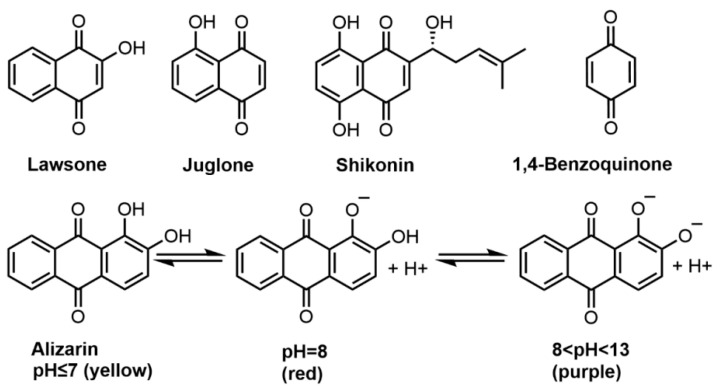
Molecular structures of lawsone, juglone, shikonin, 1,4-benzoquinone, alizarin and the color changes at acid–base conditions.

**Figure 3 molecules-27-08062-f003:**
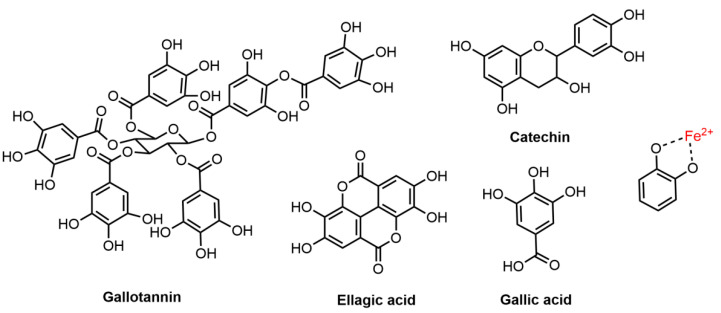
Chemical structures of tannins (gallotannin, ellagic acid, gallic acid and catechin) with a proposed mordanting mechanism when complexed with iron (II) ions.

**Figure 4 molecules-27-08062-f004:**
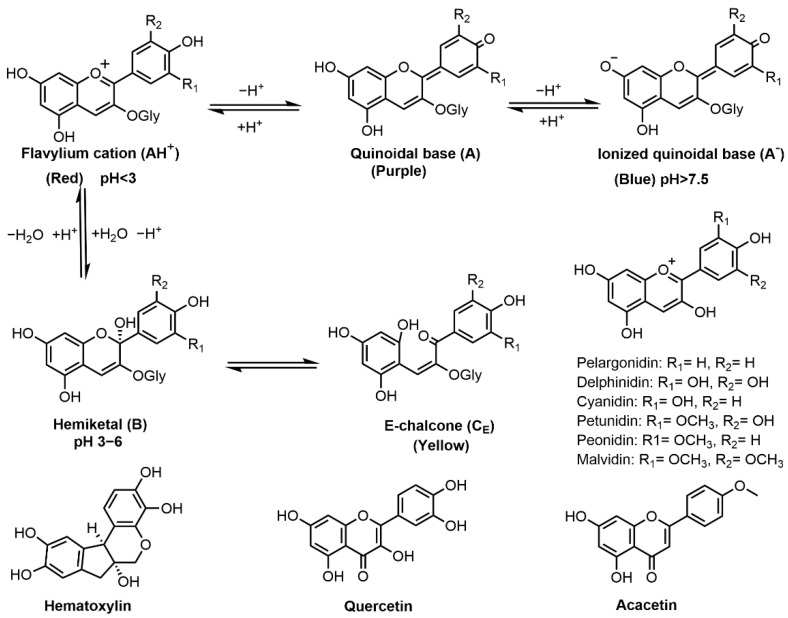
Chemical structures of flavonoids (anthocyanins, hematoxylin, quercetin and acacetin) and the effect of pH on anthocyanin structure and resultant color.

**Figure 5 molecules-27-08062-f005:**
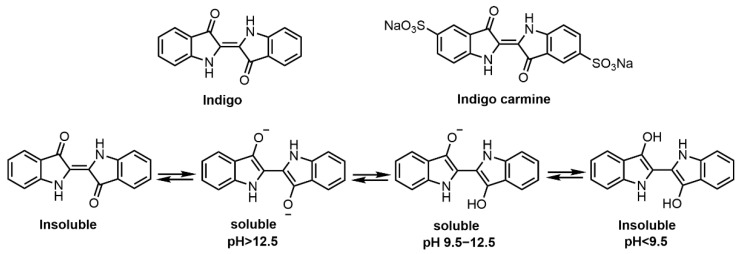
Molecular structures of indigo, indigo carmine and the pH effect on the soluble forms of indigos.

**Figure 6 molecules-27-08062-f006:**
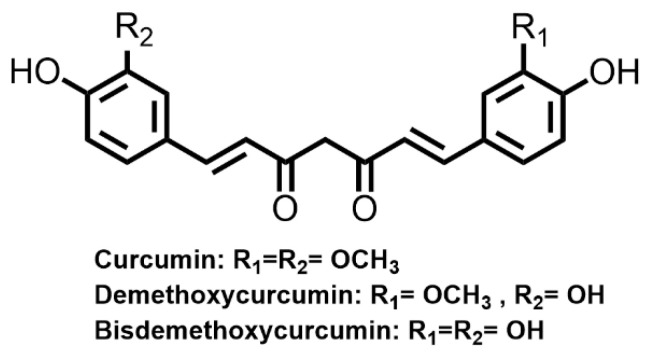
Molecular structures of the main types of curcuminoids.

**Figure 7 molecules-27-08062-f007:**
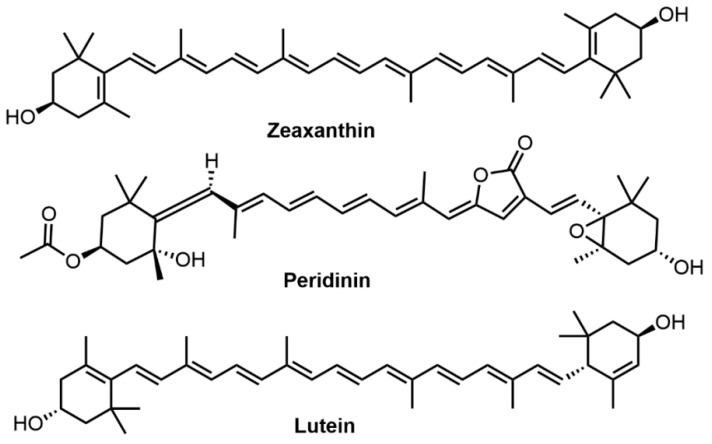
Molecular structures of zeaxanthin, peridinin, and lutein.

**Table 1 molecules-27-08062-t001:** Summary of natural plant colorants used for hair dye applications.

Category	Colorant	Botanical Origin	Extraction Process	Dye Bath	Dyeingsubstrate	Mordant	Dyeing Process	Dyed Color	Color Fastness	Refs.
Quinones	lawsone	leaves of *Lawonia inermis* L.	ultrasound reflux extraction (sodium hydroxide solution 0.25 mol/L, solid liquid ratio 1:55, 140 min, 100 °C)	dye gels (xanthan gum, 1,2-propanediol)	gray hair	iron (II) sulfate	post-mordanting dyeing	reddish brown	/	[16]
	lawsone	leaves of *Lawonia inermis* L.	reflux extraction (distilled water, solid liquid ratio 1:6, 120 min, 100 °C)	emulsion	gray hair	iron (II) sulfate	post-mordanting dyeing	blank	resistant to 20 shampoo washes	[20]
	lawsone	leaves of *Lawonia inermis* L.	ultrasound reflux extraction (sodium hydroxide 0.25 mol/L, solid liquid ratio 1:55, 140 min, 100 °C)	dye gels (xanthan gum, 1,2-propanediol)	yak hair	iron (II) sulfate	post-mordanting dyeing	reddish brown	resistant to 15 shampoo washes	[17]
	lawsone	leaves of *Lawonia inermis* L.	cold maceration extraction (water, 48 h)	dye gels (carbopol-934, glycerin, sodium hydroxide solution, methyl paraben)	hair	/	direct dyeing	brownish	resistant to 5 shampoo washes	[18]
	lawsone	leaves of *Lawonia inermis* L.	/	paste	goat hair	/	direct dyeing	reddish brown	/	[42]
	juglone	husk of *Juglans regia* L.	solvent extraction with microwave-assisted (acetone–water 70% (*v*/*v*), 60 s, 180 w) and ultrasound-assisted (20 min, 90 w, 37 kHz)	solution	bleached hair	iron (II) sulfate and *aloe vera* gel	meta-mordanting dyeing	dark brown	resistant to 15 shampoo washes	[12]
	juglone	husk of *Juglans regia* L.	ultrasound reflux extraction (ethanol 50%, solid liquid ratio 1:25, 120 min, 60 °C)	dye gels (xanthan gum, 1,2-propanediol)	gray hair	iron (II) sulfate	post-mordanting dyeing	brown	/	[16]
	juglone	husk of *Juglans regia* L.	Solvent extraction (dichloromethane, 60 min, 3 times)	solution	yak hair	/	direct dyeing	red brown	/	[11]
	shikonin	roots of *Lithospermum erythrorhizon* Sieb. et Zucc.	solvent extraction (ethanol and 3% acetic acid, 30 h)	solution	bleached hair	/	direct dyeing	light brown grey	resistant to 8 shampoo washes	[21]
	alizarin	roots of *Rubia tinctoria* L.	/	solution	yak hair	/	direct dyeing	red	/	[11]
	benzoquinone	shoots of *Pyrus lindleyi* Rehd.	solvent extraction (phosphate buffer, pH 6.0)	solution	yak hair	/	direct dyeing	brown	/	[11,43]
Tannins	gallotannin	parasitic aphids of *Rhus chinensis* Mill.	ultrasound reflux extraction (80% ethanol, solid liquid ratio 1:25, 160 min, 60 °C)	dye gels (Xanthan gum, 1,2-propanediol)	gray hair	iron (II) sulfate	post-mordanting	black	resistant to 13 shampoo washes	[44]
	gallotannin	parasitic aphids of *Rhus chinensis* Mill.	ultrasound reflux extraction (80% ethanol, solid liquid ratio 1:25, 160 min, 60 °C)	dye gels (Xanthan gum, 1,2-propanediol)	gray hair	iron (II) sulfate	post-mordanting	black	/	[44]
	catechin	matcha tea	/	solution	unpigmented hair	iron (II) lactate	post-mordanting	dove grey	resistant to 12 shampoo washes	[36]
Flavonoids	cyanidin-3-o-rutinoside	fruit skins of *Ribes nigrum* L.	aqueous extraction (acidified water, 2 h)	paste	bleached hair	/	direct dyeing	blue	resistant to 12 shampoo washes	[45]
	cyanidin-3-glucoside	fruit of *Morus nigra* L.	solvent extraction (methanol with aq. hydrochloric acid 1%, 30 min)	solution	yak hair	iron (II) oxalate	meta-mordanting	blue	/	[11]
	cyanidin-3-glucoside	beans of *Phaseolus mungo*	solvent extraction (hydrochloric ethanol, 4 °C, 24 h)	dye gels	bleached hair	/	direct dyeing	brownish red	resistant to 4 shampoo washes	[46]
	cyanidin-3-glucoside	fruit of *Cleistocalyx nervosum* Var. Paniala	solvent extraction (hydrochloric ethanol, 24 h)	spray	bleached hair	/	direct dyeing	red	resistant to 5 shampoo washes	[47]
	cyanidin-3-glucoside	corn cobs of *Zea mays L. Var.*	aqueous extraction (80 ± 2 °C, 15 min)	solution	grey hair	/	direct dyeing	blue	/	[48]
	hematoxylin	heartwood of *Haematoxylon campechianum*	ultrasound reflux extraction (ethanol 80%, solid liquid ratio1:25, 160 min, 60 °C)	dye gels (Xanthan gum, 1,2-propanediol)	gray hair	iron (II) sulfate	post-mordanting	brown red	/	[16]
	hematoxylin	heartwood of *Haematoxylon campechianum*	aqueous extraction (pH 9,25 °C, 1:4 (*w*/*v*))	solution	bleached hair	iron (II) sulfate	meta-mordanting	reddish-brown	resistant to 15 shampoo washes	[19]
	hematoxylin	heartwood of *Haematoxylon campechianum*	aqueous extraction (95 °C, 50 min, 4 times)	solution	bleached hair	/	direct dyeing	light brown red	resistant to 8 shampoo washes	[21]
	quercetin	leaves of *Tectona grandis* Linn. F.	solvent extraction (ethanol-water, 70 °C, 3 h, 2 cycles)	solution	bleached hair	/	direct dyeing	brown	/	[49]
	acacetin	bark of *Acacia farnesiana* (Linn.) Willd.	reflux extraction with (distilled water, solid liquid ratio 1:6, 120 min, 100 °C)	solution	gray hair		direct dyeing		/	[20]
Indigo	indigo	leaves of Isatis indigotica Fort., Polygonum tinctorium Ait.	/	paste	gray hair		direct dyeing	dark brown	Resistant to 6 shampoo washes	[50]
	indigo carmine	/	/	solution	blonde hair	/	direct dyeing	blue	/	[51]
Curcuminoids	curcumin	root of *Curcuma longa* Linn.	aqueous extraction (4 °C, pH 5, solid liquid ratio 1:4)	solution	bleached hair	iron (II) sulfate	meta-mordanting	orangish brown	resistant to 15 shampoo washes	[19]
	curcumin	root of *Curcuma longa* Linn.	/	solution	yak hair	/	direct dyeing	yellow	/	[11,52]
Carotenoids	zeaxanthin	aerial parts of *Eclipta alba* L.	aqueous extraction (100 °C, pH 9, solid liquid ratio 1:4)	solution	bleached hair	iron (II) sulfate	meta-mordanting	brown	resistant to 15 shampoo washes	[19]
	peridinin	fruit of *Terminalia belerica* Roxb.	aqueous extraction (25 °C, pH 7, solid liquid ratio 1:4)	solution	bleached hair	iron (II) sulfate	meta-mordanting	brown	resistant to 15 shampoo washes	[19]
	lutein	flower of *Tagetes erecta* Linn.	aqueous extraction (60 min, 100 °C)	solution	grey hair	*aloe vera* gel	meta-mordanting	black	resistant to 5 shampoo washes	[40]

## Data Availability

Not applicable.

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
