# Peer review of "Recent Advancements in Natural Plant Colorants Used for Hair Dye Applications: A Review"

_molecules, 2022, doi:10.3390/molecules27228062_

Round 1

Reviewer 1 Report

Reviewer #1:

This work systematically reviews recent advance in natural plant colorants for hair dye applications. After go through this work, I found this is a well-written manuscript. It is significative and I would like to suggest the publication of this work after some minor revision.

1. The representative hair dying effects obtained from the different types of natural hair dyes are suggested to be added in the corresponding Figures.

2. The recently developed hair dying methods using polydopamine and its derivative as the melanin-mimicking pigments are suggested to be added in the revised manuscript, and the relevant references such as ACS Central Science, 2020, 6(7): 1179-1188; RSC Adv., 2019, 9, 33617–33624.

Author Response

Reviewer #1

This work systematically reviews recent advance in natural plant colorants for hair dye applications. After go through this work, I found this is a well-written manuscript. It is significative and I would like to suggest the publication of this work after some minor revision.

RESPONSE: Thank you very much for your positive comments.

  1. The representative hair dying effects obtained from the different types of natural hair dyes are suggested to be added in the corresponding Figures.

RESPONSE: Thank you for this comment. The dyeing effects of plant colorants can be variable depending on the colorants, dyeing substrate, mordanting agents, pH and the use of other developers in the dyeing process. Therefore, the dyeing effects (dyed hair color and fastness properties) of different natural plant colorants were detailed in Table 1 together with their botanical origin, extraction process, dye bath, dyeing substrate, mordant type and dyeing process.

  1. The recently developed hair dying methods using polydopamine and its derivative as the melanin-mimicking pigments are suggested to be added in the revised manuscript, and the relevant references such as ACS Central Science, 2020, 6(7): 1179-1188; RSC Adv., 2019, 9, 33617–33624.

RESPONSE: Thank you for this comment. We have added this information on hair dying methods using polydopamine and its derivative as the melanin-mimicking pigments hair on line 437 on page 11: “Besides, oxidation of dopamine to eumelanin-like polydopamine and deposition on the surface of hair in the form of nanoparticles have shown as a novel biomimetic strategy to develop melanin-mimicking pigments for hair dyeing [112,113].”

Reviewer 2 Report

General: The advantages of natural plant colorants and their lower toxicology in comparison to synthetic hair dyes are a main theme of the paper. Also, one molecule of the latter is mentioned – p-phenylenediamine. But for a comparison, one should describe the molecules used in synthetic hair dyes in more detail – the molecule class, the coupler (such as resorcinol) and the following oxidation. Then, one has an impression, why natural hair dyes can be less toxic and more environmentally friendly. In the paper, often metal salts/metal mordants are mentioned as questionable from a toxicological point of view, but they are not used in modern oxidative hair colors, as exothermic processes would be detrimental for the hair. Summarizing, modern oxidative hair color treatment should be described and the toxicological assessment has to be improved.

 Line 173 ff.

Juglone and its use as hair dye is mentioned, but it is described as low-toxic source of natural colorants.

There has been an intense discussion about the effects of juglone, in particular because in hair dye formulations a significant amount has to be used. Also, NIH and SCCP discussed if juglone is potentially toxic and carcinogenic. SCCP opinion was that juglans regia cannot considered to be safe for hair dyeing purposes. The discussion on the toxicology of juglone should be mentioned, either here or in chapter 5, toxicological assessments.

 Line 275 ff.

Anthocyanins are presented as successful hair dye, and for prevention of a number of diseases. There arise two questions: First, how resistant are anthocyanins to washing – a general problem of this class of dyes. The authors should give a reference to this problem. Second, ref. 71 describes the use of anthocyanines in food and nutrition, which is rather far from the use as a hair dye.

 Line 285 ff.

Hematoxylin is not derived from Caesalpinia sappan (sappanwood). In the cited paper [ref. 72], however, the correct origin of Hematoxylin is given in the Introduction: "Haematoxylin is a long-known natural compound extracted from the plant Haematoxylon campechianum (bloodwood or blackwood tree)."

 Line 336, concerning Indigo: “The only hindrance to its commercial use in hair dye cosmetics is …” This should be reformulated. Indigo is widely used in commercial plant based hair colors.

 Line 450 ff.

The chapter 5. Toxicological Assessments should be completely rewritten and worked out in more detail. There are not only some errors in this part, but also important information is missing. Some examples are mentioned below:

 Heavy metals and ethanol are named in one sentence that they may cause sensitization (line 453). But one should not throw everything into the same pot. Ethanol does not cause sensitization. Moreover, in the cited ref. 14, one finds in the introduction a sentence that “High doses of heavy metals and some irritating ingredients (eg, benzyl alcohol) have been previously reported as mordant or developer in the dye bath or hair dye formulas” and a final sentence in the conclusions that “further investigations are required … to assess other toxicities (including skin sensitization of these plant hair dyes”. But nothing is found in the given reference on sensitization of ethanol or heavy metals.

 Another example: It is written that because of the skin irritation and sensitization, safety assessments of both natural colorant ingredients and hair dye formulations are compulsory. All this is already the case: Safety assessment is required by the cosmetics legislative of the different countries.

 A final example: The problematic of metallic mordants and later use of oxidative hair colors is completely missing in the toxicological assessments. This is, however, a big issue in praxi.

This should be also mentioned in the Concluding Remarks, as it is very important. The same holds for juglone, where the discussion on the toxicology is missing.

 Line 485

This is unprecise and not the case here: Alkaline formulation is NOT required for hair dyeing with plant based dyes, only for oxidative hair colors.

Author Response

Reviewer #2

General: The advantages of natural plant colorants and their lower toxicology in comparison to synthetic hair dyes are a main theme of the paper. Also, one molecule of the latter is mentioned – p-phenylenediamine. But for a comparison, one should describe the molecules used in synthetic hair dyes in more detail – the molecule class, the coupler (such as resorcinol) and the following oxidation. Then, one has an impression, why natural hair dyes can be less toxic and more environmentally friendly. In the paper, often metal salts/metal mordants are mentioned as questionable from a toxicological point of view, but they are not used in modern oxidative hair colors, as exothermic processes would be detrimental for the hair. Summarizing, modern oxidative hair color treatment should be described and the toxicological assessment has to be improved.

RESPONSE: Thank you for this constructive advice. We have added more information about the oxidative and non-oxidative hair dye types in the Introduction section on Line 29, Page 1: “Hair dye products can be grouped into three categories according to wash fastness: temporary, semi-permanent and permanent hair dyes [1]. Permanent hair dyes are referring to synthetic oxidative hair dyes, by which colors are produced in the hair cortex from small primary intermediates (e.g., p-phenylenediamine and p-aminophenol) and couplers (e.g., m-aminophenol, m-hydroxyphenol and resorcinol) through oxidation reactions in the presence of hydrogen peroxide as the oxidizing agent [2].” and on Line 39, Page 1: “By contrast, natural dyes are temporary or semi-permanent non-oxidative hair dyes that can be adsorbed onto the cuticle and some parts of the cortex of the hair shaft to produce color.” The Toxicological Assessments section has also been revised as suggested. Thank you so much.

- Line 173 ff.

Juglone and its use as hair dye is mentioned, but it is described as low-toxic source of natural colorants.

There has been an intense discussion about the effects of juglone, in particular because in hair dye formulations a significant amount has to be used. Also, NIH and SCCP discussed if juglone is potentially toxic and carcinogenic. SCCP opinion was that juglans regia cannot considered to be safe for hair dyeing purposes. The discussion on the toxicology of juglone should be mentioned, either here or in chapter 5, toxicological assessments.

RESPONSE: Thanks for this comment. Sorry, we didn’t find the SCCP opinion on juglone or walnut extract but we did find articles describing the cosmetic applications of walnut green husk extract (it is also listed in the International Nomenclature of Cosmetic Ingredients names) and cytotoxic mechanisms of juglone. We have added more application information about juglone and walnut husk extracts on Line 180, Page 4: “Juglone and juglone-containing walnut green husk extracts has been used in skin coloring preparations [52], hair dyes [11,12,16] and antimicrobial agents [53].” The cytotoxicity of juglone was also discussed on Line 187, Page 4: “Additionally, some studies reported dose-dependent cytotoxicity of juglone in human fibroblasts and keratinocytes [54,55]. These findings indicate that cosmetic preparations containing juglone should be used with care.” Also, the toxicity of juglone has also been mentioned in section 5 Toxicological Assessments.

- Line 275 ff.

Anthocyanins are presented as successful hair dye, and for prevention of a number of diseases. There arise two questions: First, how resistant are anthocyanins to washing – a general problem of this class of dyes. The authors should give a reference to this problem. Second, ref. 71 describes the use of anthocyanines in food and nutrition, which is rather far from the use as a hair dye.

RESPONSE: Thank you for this constructive advice. We have deleted the sentence “In addition to being used as natural colorants, anthocyanins are well known for their anti-diabetes, anti-cancer, anti-inflammatory, antibacterial and anti-obesity activities and prevention of cardiovascular diseases [71].” and added a description of its washing fastness on Line 284, Page 7: “However, anthocyanins are instable in aqueous formula whereas the addition of vitamin E acetate at 0.04% can enhance this natural colorant’s stability [75]. Anthocyanins-based preparations have shown to be efficient semi-permanent dyestuffs for hair with the dyed colors durable up to 5 wash cycles [76].”

- Line 285 ff.

Hematoxylin is not derived from Caesalpinia sappan (sappanwood). In the cited paper [ref. 72], however, the correct origin of Hematoxylin is given in the Introduction: "Haematoxylin … is a long-known natural compound extracted from the plant Haematoxylon campechianum (bloodwood or blackwood tree)."

RESPONSE: Thanks for this comment. The Latin name of “Haematoxylon campechianu” has been used throughout the article.

- Line 336, concerning Indigo: “The only hindrance to its commercial use in hair dye cosmetics is …” This should be reformulated. Indigo is widely used in commercial plant based hair colors.

 RESPONSE: Thanks for this comment. We have revised the text on line 338, page 8  to “Overall, indigo is a sustainable and environmentally friendly natural colorant and has been widely used in commercial hair dye cosmetics with henna for dark colors.”.

- Line 450 ff.

The chapter 5. Toxicological Assessments should be completely rewritten and worked out in more detail. There are not only some errors in this part, but also important information is missing. Some examples are mentioned below:

Heavy metals and ethanol are named in one sentence that they may cause sensitization (line 453). But one should not throw everything into the same pot. Ethanol does not cause sensitization. Moreover, in the cited ref. 14, one finds in the introduction a sentence that “High doses of heavy metals and some irritating ingredients (eg, benzyl alcohol) have been previously reported as mordant or developer in the dye bath or hair dye formulas” and a final sentence in the conclusions that “further investigations are required … to assess other toxicities (including skin sensitization of these plant hair dyes”. But nothing is found in the given reference on sensitization of ethanol or heavy metals.

Another example: It is written that because of the skin irritation and sensitization, safety assessments of both natural colorant ingredients and hair dye formulations are compulsory. All this is already the case: Safety assessment is required by the cosmetics legislative of the different countries.

A final example: The problematic of metallic mordants and later use of oxidative hair colors is completely missing in the toxicological assessments. This is, however, a big issue in praxi.

This should be also mentioned in the Concluding Remarks, as it is very important. The same holds for juglone, where the discussion on the toxicology is missing.

RESPONSE: Thank you for your constructive advices. The first paragraph of “toxicological assessments " section has been rewritten as suggested.

Line 450, Page 11, rephrase: “Natural origin does not necessarily mean non-toxic or safe. Black henna, a combination of henna leaves and p-phenylenediamine (PPD), has been known to cause allergic contact dermatitis [45,116]. Some colorants, such as juglone, are natural toxins with cyto-toxic effects [54,55]. Also, the colorant contents and toxic effects of plant extracts used in hair dye preparations can be variable with different plant sources, harvest seasons and extraction techniques. The application compatibility of juglone-containing walnut husk extract as a hair dye ingredient has been assessed both in vitro and in vivo, showing it is not irritating to the skin [12,16]. Besides colorants, metallic mordants and heavy metal contaminants from plant extracts can be accumulated in humans with frequent dyeing and may pose health impacts. For example, the presence of high levels of Fenton transition metals (e.g. iron) can cause photooxidative damage of hair fiber [38]. Heavy metals, such as cadmium, chromium and lead, can be absorbed in the skin, liver and kidney, causing allergic contact dermatitis [117], reproductive system dysfunction [118], and other adverse health effects. Several novel analytical methods, such as differential pulse anodic stripping voltammetry, have been developed for the rapid detection of heavy metals [119,120]. Therefore, safety assessments of both natural colorant ingredients and hair dye formulations are compulsory. It is also necessary to establish toxicological database of hair dye plants and their main colorants since currently most hair dye plant extracts used in cosmetics are lacking important toxicological information such as acute/sub-chronic systemic toxicity, irritancy, allergenicity and genotoxicity [121].”

- Line 485

This is unprecise and not the case here: Alkaline formulation is NOT required for hair dyeing with plant based dyes, only for oxidative hair colors.  

RESPONSE: Thank you for this comment. The text “Instability in alkaline formulation required for hair dyeing” has now been removed from section 6. Concluding Remarks.

Reviewer 3 Report

An interesting article on natural hair dyes is offered, discussing their advantages and disadvantages.
I have two recommendations:
The introduction should be expanded by introducing modern techniques for creating natural dyes;
The goal is to refine it in order to make it sound clearer;

Author Response

Reviewer #3

An interesting article on natural hair dyes is offered, discussing their advantages and disadvantages. I have two recommendations:

RESPONSE: Thank you very much for your positive comments.

-The introduction should be expanded by introducing modern techniques for creating natural dyes;

-The goal is to refine it in order to make it sound clearer;

RESPONSE: Thank you for your constructive advice. We have added the production of modern natural colorants in the Introduction section on Line 63, Page 2: “Natural dyes used in commercial cosmetics are mainly extracted from plants by sol-vent extraction [19], ultrasonic assisted extraction [22], microwave assisted extraction [10], supercritical fluid extraction [23], enzyme-assisted extraction [24] etc.” In addition, we reviewed the use of novel synthetic biology techniques in chapter 4.1 for natural colorant production on Lines 388-409, Page 10.

Author Response

Reviewer #4

This is a highly topical area of interest in colour research, not well served by other reviews. The content is described well in the abstract and the introduction, and a good justification is given for its timeliness based on ‘green’ and toxicological considerations. The coverage is appropriate, comprehensive, and extensively referenced, so that the manuscript provides a sound basis for a publishable review in this area. There is a good balance between historical and current contents. I have some concerns on the scientific presentation, arguments, and explanations, especially in the

earlier parts of the article. Some of these might be addressed by the serious improvement in the English usage, including technical terminology, and style that is required throughout the article. Specific comments are as follows.

RESPONSE: Thank you very much for your positive comments. The revised manuscript has been carefully and thoroughly proofread.

Introduction:

  1. PPD is a hair dye precursor or component not in itself a hair dye.

RESPONSE: Thank you for this constructive comment. We have rephrased the text on Line 29, Page 1 with more details about oxidative hair dyes: “Hair dye products can be grouped into three categories according to wash fastness: temporary, semi-permanent and permanent hair dyes [1]. Permanent hair dyes are referring to synthetic oxidative hair dyes, by which colors are produced in the hair cortex from small primary intermediates (e.g., p-phenylenediamine and p-aminophenol) and couplers (e.g., m-aminophenol, m-hydroxyphenol and resorcinol) through oxidation reactions in the presence of hydrogen peroxide as the oxidizing agent [2].”

  1. ‘Potassium solution’ – of what? Potassium alone is not a bleaching agent so presumably there are other ingredients.

RESPONSE: Thank you for this comment. The “potassium solution” has been revised as “potassium lye solution” on Line 54, Page 2.

Hair coloring mechanisms

A methyl group is not normally reactive

RESPONSE: Thank you for this comment. The “reactive methyl” has been removed on Line 76, Page 2.

  • Iron (II) is preferable to ferrous throughout the article. Also, copper (II)

RESPONSE: Thank you for this comment. We have replaced ferrous with iron (II) and cupric with copper (II) throughout the article.

  1. Use ‘Dative covalent bonding’ throughout.

RESPONSE: Thank you for this comment. The text on Line 113, Page 3, has been corrected as “Dative covalent bonding” as suggested.

  1. Fig 1(b) shows dative covalent bonding between Fe2+ and the polyphenol, which bonds to the protein by hydrogen bonding.

RESPONSE: The text on Line 115, Page 3, has been corrected as “Fig 1(b) shows dative covalent bonding between iron (II) ions and a polyphenol dye, which bonds to the protein by hydrogen bonding [36].”. Thank you very much. 

  1. I question the link between ‘heavy metals’ and the Fenton reaction. The Fenton reaction usually involves iron, which is not usually regarded as a heavy metal.

RESPONSE: The text on Line 121, Page 3, has been revised as: “the treatment of transition metal mordants results in the accumulation of iron and copper in human hairs …”.

  • ‘Transitional’ metal should be ‘transition’ metal

RESPONSE: Thank you for this comment. “Transitional” has been replaced with “transition” in the revised manuscript as suggested.

  • Figure 1 caption. ‘mesomeric’ does not appear to be the correct term here.

RESPONSE: Thank you for this comment. We have revised the figure caption on Line 134, Page3, as: “Possible interactions between hair fiber, polyphenol dye and mordanting iron (II) ions, where R= H, CH3, OH [36] (b).”

  1. R = H, CH3, OH

RESPONSE: Thank you for pointing this out. “R = H, CH3, OH” has been corrected in the figure caption as suggested.

  1. There is no ferrous lactate in this diagram.

RESPONSE: We have replaced “ferrous lactate” with “mordanting iron (II) ions” in the figure caption. Thank you.

Phytochemicals

  1. ‘Affluent’ does not appear to be the correct word used with phenolic….. (several?)

RESPONSE: The “affluent” has been replaced with “several” on Line 229, Page 6, as suggested. Thank you.

  1. ‘portentous’ is also unusual in this context. (important?)

RESPONSE: Thank you for pointing out a misspelling. It has now been corrected to “protein” on Line 255, Page 6. 

Round 2

Reviewer 2 Report

The toxological assessments are still rather short, but as they are not the focus, but only a small part of the paper, this is ok. The paper is sufficiently improved for publication.

Reviewer 4 Report

Authors have corrected the paper according to my comments